# Diagnosis of Cholangiocarcinoma

**DOI:** 10.3390/diagnostics13020233

**Published:** 2023-01-08

**Authors:** Dong Woo Shin, Sung-Hoon Moon, Jong Hyeok Kim

**Affiliations:** 1Department of Internal Medicine, Hallym University College of Medicine, Hallym University Sacred Heart Hospital, Gyeonggi-do, Anyang 14068, Republic of Korea; 2Department of Internal Medicine, Chungang University College of Medicine, Chungang University Gwangmyeong Hospital, Gyeonggi-do, Gwangmyeong 14353, Republic of Korea

**Keywords:** cholangiocarcinoma, diagnosis

## Abstract

Cholangiocarcinoma (CCA), a tumor of the bile duct epithelium, is increasing in incidence. CCA remains a highly fatal malignancy because early diagnosis is difficult. Based on its anatomical location, CCA can be categorized into the following three groups: perihilar, intrahepatic, and extrahepatic. Patients with CCA complain of asymptomatic jaundice, weight loss, and right upper quadrant abdominal discomfort. Imaging modalities, including transabdominal ultrasound, computed tomography, and magnetic resonance imaging, play an important role in detecting tumors as well as guiding biopsy procedures and staging workups in CCA. Characteristically, extrahepatic CCA shows abrupt changes in ductal diameter with upstream ductal dilation. Endoscopic ultrasound (EUS) and endoscopic retrograde cholangiopancreatography (ERCP) are recommended as the next step in the evaluation of extrahepatic CCA. Tissue is obtained through EUS-FNA or ERCP (biopsy, brush cytology), and therapeutic intervention (such as stent insertion) is performed with ERCP. Moreover, several serum tumor markers (carbohydrate antigen 19-9 and carcinoembryonic antigen) can be useful in diagnosing CCA in some patients.

## 1. Introduction

Cholangiocarcinoma (CCA), a malignant tumor that arises from the bile duct epithelium, is the second most prevalent type of liver cancer, following hepatocellular carcinoma (HCC). Anatomically, CCA can be categorized into the following three groups: perihilar, intrahepatic, and extrahepatic [1]. Perihilar CCA (or Klatskin tumor) is the most prevalent type, comprising 50–60% of CCA, followed by extrahepatic CCA (20–30%) and intrahepatic CCA (10–20%) [1,2]. Perihilar CCA is a tumor that grows between the second branch of the intrahepatic duct and the cystic duct [3]. Extrahepatic CCA originates from the bile duct between the ampulla of Vater and the cystic duct [4]. Intrahepatic CCA arises from the tiny bile ducts distal to the secondary branches of the intrahepatic ducts [5].

The incidence of CCA is less than 6 cases per 100,000 people globally [6], yet the regional differences are significant. In several Asian nations (South Korea, Thailand, and China), the incidence of CCA is substantially greater (>6 cases per 100,000 people) compared to the Western countries (0.35–2 cases per 100,000 people) [6]. During the past few decades, the incidence of CCA has grown in many countries. [7,8]. The intrahepatic CCA has especially increased in incidence by 350%, while the extrahepatic CCA has increased by 20%. [2]. The established risk factors for CCA include primary sclerosing cholangitis, parasite infection, biliary-duct cysts, and hepatolithiasis. Less-established risk factors include inflammatory bowel disease, chronic liver disease (cirrhosis, chronic hepatitis B and C), obesity, diabetes, alcohol, smoking, and genetic polymorphisms [6,9].

Despite the advances in the diagnosis and treatment of CCA, the survival rates remain poor. Population-based research in the United States revealed that the median overall survival for extrahepatic and intrahepatic CCA was 8 and 4 months, respectively, between 1973 and 2008 [2]. In accordance with these results, nationwide research in Thailand from 2009 to 2013 revealed a high, constant 1-year mortality rate of 81.7% [10]. CCA is considered a highly fatal malignancy for the following reasons: (i) biological aggressiveness; (ii) locally advanced state at presentation; (iii) a high recurrence rate following therapy.

## 2. Clinical Presentation

### 2.1. Signs and Symptoms

Jaundice, dark urine, clay-colored stool, and pruritus can occur when extrahepatic CCA obstructs the bile duct. Patients with CCAs involving only the intrahepatic ducts rarely develop jaundice but may experience right upper quadrant abdominal pain (dull, continuous), malaise, night sweats, weight loss, or cachexia [11]. In some cases, CCA is discovered incidentally during workup for abnormal liver blood tests, even in asymptomatic patients [12].

### 2.2. Physical Examination

Physical examination findings in patients with extrahepatic CCA include jaundice, hepatomegaly, and palpable mass in the right upper quadrant area [13]. In contrast, patients with intrahepatic CCA usually have no symptoms other than right upper quadrant tenderness. The Courvoisier sign (a palpable gallbladder in a jaundiced patient) is considered a sign of pancreaticobiliary malignancy. These findings, however, are limited because they can also be observed in other diseases, such as chronic pancreatitis, biliary obstruction, and choledochal cyst [14]. Patients with CCA may have rare cutaneous manifestations of paraneoplastic syndromes such as Sweet syndrome [15], erythema multiforme [16], or porphyria cutanea tarda [17].

## 3. Laboratory Findings

Total and direct bilirubin, gamma-glutamyltransferase (GGT), alkaline phosphatase (ALP), and serum transaminase (aspartate aminotransferase [AST] and alanine aminotransferase [ALT]) should be checked on all patients with suspected CCA to identify the presence of cholestasis. Total bilirubin (generally exceeding 10 mg/dL), direct bilirubin, ALP, and GGT are typically elevated in patients with extrahepatic CCA (usually increased 2- to 10-fold). Initially, AST and ALT levels may be normal, but hepatocellular damage caused by persistent cholestasis leads to an increase in transaminases and a prolonged prothrombin time. Patients with intrahepatic CCA usually have abnormal ALP values, but serum bilirubin levels are typically mildly raised or normal [13].

The key blood biomarkers for the diagnosis of CCA are carbohydrate antigen 19-9 (CA 19-9) and carcinoembryonic antigen (CEA). However, its weakness as a diagnostic marker lies in the limited sensitivity for detecting early-stage CCA and the possibility that it could be raised in benign diseases. Alpha-fetoprotein (AFP) can be used to differentiate intrahepatic CCA from hepatocellular carcinoma (HCC).

### 3.1. Carbohydrate Antigen 19-9

CA 19-9 is an epitope of sialyl-Lewis antigens produced by the pancreaticobiliary duct, salivary, endometrial, gastric, and colonic epithelial cells [18]. CA 19-9 is an established blood biomarker for the diagnosis of CCA, with a sensitivity of 50–90 percent and a specificity of 54–98 percent [19,20,21]. Elevated CA 19-9 at presentation is related to poor prognosis [22,23], and considerable rises in CA 19-9 concentrations (>1,000 units/mL) are indicative of unresectable disease [20,24]. Moreover, the serum levels of CA 19-9 can be used to identify CCA in patients with primary sclerosing cholangitis [25].

However, there are several problems in using serum CA 19-9 as a diagnostic biomarker for CCA. First of all, CA 19-9 has a limited specificity since it is frequently increased in patients with benign diseases (cholangitis or bile duct stricture). Secondly, the cutoff value of CA19-9 used to differentiate malignant from benign biliary tract diseases differs depending on whether cholangitis or cholestasis is present [26,27]. Lastly, in Lewis-antigen-deficient patients (an estimated 5–10 percent of the population), CA 19-9 does not increase even in the presence of CCA [27].

### 3.2. Carcinoembryonic Antigen

CEA is a glycoprotein associated with the cell membrane whose expression differs between normal tissues and malignant cells. Serum levels of CEA may be elevated in patients with CCA. In the diagnosis of CCA, a blood CEA level greater than 5.2 ng/mL showed a sensitivity of 68% and a specificity of 82% [28]. Serum CEA is not sensitive nor specific enough to identify CCA on its own since it can be elevated in benign illnesses, such as peptic ulcer disease, gastritis, diverticulitis, and liver disease, in addition to various primary gastrointestinal cancers. CEA can be used to assess the effect of treatment and to identify disease recurrence when CA19-9 at presentation is not elevated.

### 3.3. Alpha-Fetoprotein

Serum AFP levels should be evaluated in every patient with a solid liver lesion. Serum AFP is widely used in the diagnosis of HCC and germ cell tumors [29]. A series of studies from a Japanese liver cancer research team showed that 19% of patients with intrahepatic CCA had a serum AFP level greater than 20 ng/mL, 10.3% had greater than 200 ng/mL, and only 6.3% had greater than 1000 ng/mL [30]. AFP has a high specificity for identifying HCC but a low sensitivity and specificity for CCA.

### 3.4. Serum IgG4

CCA can be mistaken for IgG4-associated sclerosing cholangitis. In patients with suspected CCA, the importance of IgG4 testing is unclear. Serum IgG4 levels can be measured if IgG4-related sclerosing cholangitis is suspected. However, serum IgG4 levels can also be increased in patients with CCA [31].

## 4. Radiologic Findings

### 4.1. Transabdominal Ultrasound

Transabdominal ultrasound (TAUS) is the most commonly used initial imaging modality for jaundiced patients to confirm the presence of bile duct dilatation, identify the cause of the obstruction, and rule out gallstones [32]. In conditions when TAUS cannot identify a benign etiology of biliary obstruction, cross-sectional imaging (CT or MRI), endoscopic ultrasound (EUS), or endoscopic retrograde cholangiopancreatography (ERCP) are essential for confirming the diagnosis. TAUS has a high sensitivity for detecting biliary tract dilatation and evaluating the degree of obstruction. In a study involving 429 patients with obstructive jaundice over a 10-year period, the sensitivity of detecting ductal obstruction was 89%, and the sensitivity of localizing the site of obstruction was 94% [33]. TAUS is the best choice for patients with right upper abdominal pain without jaundice because the sensitivity of detecting gallstones or biliary dilatation is superior to that of CT. However, when the possibility of CCA is high, it is better to select cross-sectional imaging rather than TAUS. The TAUS findings of various types of CCA are as follows [34]: (1) hilar CCA (Klatskin tumor) is characterized by bilateral intrahepatic duct nonunion and segmental dilatation; (2) papillary tumors can be observed as polypoid masses inside the biliary tract; (3) nodular CCAs are characterized as isolated, smooth masses with mural thickening; (4) intrahepatic CCA is detected as a large mass with irregular margins. The tumor can be predominately hypo- or hyperechoic or have mixed echogenicity depending on the amount of internal fibrosis, mucin, and calcification. In the absence of stones, ductal dilatation (greater than 6 mm in normal people with an intact gallbladder) suggests a biliary obstructive lesion. Proximal extrahepatic CCAs may only induce dilatation of the intrahepatic ducts, whereas more distant CCAs cause dilation of both intrahepatic and extrahepatic ducts [35]. Tumors are detected at sites of abrupt ductal diameter changes. The limitation of TAUS is that the distal common bile duct may be masked by duodenal air. Therefore, biliary dilatation is frequently used as a surrogate indicator of distal biliary obstruction. If underlying PSC or cirrhosis is present, bile duct dilatation may not be observed in CCA. Nevertheless, if there is progressive biliary dilatation in the setting of a dominant stricture in PSC patients, CCA should be strongly suspected [36]. Duplex ultrasound can also be used to evaluate vascular involvement (i.e., encasement, compression, or thrombosis of the portal vein or the hepatic artery).

### 4.2. Multidetector Computed Tomography

In patients with suspected CCA, a multidetector-row CT (MDCT) scan is widely used as an alternative to TAUS due to its broad availability. It is helpful for finding intrahepatic malignancies, determining the extent of biliary stenosis, and detecting liver atrophy. MDCT can also aid in differentiating malignant from benign intrahepatic bile duct strictures (especially during the portal venous phase) and reveal nodal status [37,38].

#### 4.2.1. Intrahepatic Cholangiocarcinoma

There are the following three types of intrahepatic CCA tumor growth: periductal-infiltrating, mass-forming, and mixed. The mass-forming type is the most prevalent, comprising roughly 60% of all intrahepatic CCAs. In contrast, the periductal type and the mixed type each comprise about 20% [31]. When intrahepatic lesions are discovered in patients without liver cirrhosis, metastases from another organ should be considered as the primary diagnosis. In cirrhotic patients, the subsequent diagnostic step is to distinguish between CCA and HCC. Intrahepatic CCA is typically observed as a well-defined or infiltrative hypodense hepatic lesion with bile duct dilatation on MDCT. In around 20% of the cases, the dense fibrotic structure of the tumor may cause capsular retraction. Intrahepatic CCA typically exhibits a peripheral rim enhancement in both the arterial and venous phases [37,39]. Similar to HCC, small mass-forming intrahepatic CCAs may exhibit hyperenhancements during the arterial phase and demonstrate washout during the delayed venous phase [40].

On MDCT, the imaging characteristics of combined hepatocellular-cholangiocellular carcinoma (HCC-CCA) may differ from those of HCC and CCA. Typical enhancing patterns or biliary ductal dilatation that can differentiate between HCC and CCA are not always observed [41,42]. These mixed tumors are staged as intrahepatic CCAs rather than hepatocellular malignancies. In the lack of typical characteristics suggestive of HCC or CCA, a biopsy is required for pathological confirmation.

#### 4.2.2. Extrahepatic Cholangiocarcinoma

In patients with extrahepatic and perihilar CCA, the level of ductal dilatation reflects the location of the obstructive lesion. Perihilar CCA is suggested when both intrahepatic ducts are dilated, and the left and hepatic ducts are separated. Because intrahepatic CCA does not develop jaundice until later in the disease course, it tends to become huge and invade the adjacent liver parenchyma. Intrahepatic ductal dilatation and hepatic lobar atrophy may suggest the site of origin.

The dilatation of the intrahepatic and extrahepatic duct, as well as the gallbladder suggests a periampullary tumor. An enlarged gallbladder without dilated intrahepatic or extrahepatic ducts is suggestive of cystic duct malignancy or stones.

### 4.3. MRI and MRCP

On MRI, CCAs present as hypointense lesions on T1-weighted images and as heterogeneously hyperintense lesions on T2-weighted images [43]. Central hypointensity on T2-weighted imaging may suggest intra-tumoral fibrosis. The tumor shows gradual and concentric filling after peripheral enhancement upon contrast injection. Contrast enhancement on delayed imaging suggests a peripheral CCA.

The appearance of mixed HCC-CCA on MRI is distinct [44,45,46]. On gadoxetic acid-enhanced MRI, a strong enhancing rim and irregular margin are suggestive of a mixed HCC-CCA, whereas lobulated shape, target appearance, and weak enhancing rim are suggestive of a mass-forming intrahepatic CCA [45]. Mixed HCC-CCA and atypical hypovascular HCC can be distinguished by the target appearance [46]. MRI with MRCP is a noninvasive method for evaluating the biliary tract and can be performed as a preoperative evaluation of CCA since MRCP does not require the injection of contrast material into the bile duct, unlike ERCP. Moreover, MRCP has several advantages over MDCT. In addition to the ability to scan intrahepatic disorders, it enables the construction of a three-dimensional image of the bile duct and other vascular structures.

MRCP provides a variety of information on disease extent and resectability comparable to MDCT, ERCP, and angiography [47,48,49,50]. In a study comparing MRCP with ERCP in 40 patients with perihilar CCA, both techniques identified 100% of biliary obstructions. However, MRCP was superior in determining the anatomical extent of the malignancy and the underlying cause of jaundice [51].

In one study, a combination of MRCP and MDCT could replace invasive cholangiography in patients with obstructive jaundice due to proximal lesions [52]. However, interventions such as biopsy, stone extraction, or stent insertion cannot be performed with MRCP. If possible, MRCP should be considered prior to biliary drainage since it is difficult to evaluate a collapsed bile duct after biliary drainage.

### 4.4. PET Scan

Positron emission tomography with fluorodeoxyglucose (FDG-PET) scanning is widely used for CCA staging. PET scans and combined PET/CT scans are not superior to MDCT or MRI with MRCP scans for the diagnosis of the primary tumor; nonetheless, PET scans are useful for detecting occult metastases. Therefore, a PET scan is performed in patients with potentially resectable CCA before surgical resection.

CCAs are detectable on FDG-PET due to the high glucose uptake of the bile duct epithelium. Most CCAs are (18)F-fluorodeoxyglucose avid tumors [53]. PET and integrated PET/CT can detect nodular CCAs as small as one centimeter, but they are less effective for infiltrating tumors, which may not accumulate FDG [54,55]. A semiquantitative assessment of the maximum standardized uptake value (SUVmax) and tumor-to-normal liver (T/N) ratio could be used to distinguish malignant from benign lesions [56,57]. However, the appropriate cutoff value for distinguishing benign from malignant lesions has not yet been determined.

### 4.5. Chest CT

Consensus-based guidelines from the National Comprehensive Cancer Network (NCCN) recommend chest CT for potential surgical candidates during the initial workup of CCA [58].

## 5. Endoscopic Findings

### 5.1. Cholangiography

Cholangiography is a procedure used to detect abnormalities by injecting contrast material into the biliary tract via percutaneous (percutaneous transhepatic cholangiography [PTC]) or endoscopic (ERCP) routes. Cholangiography can be used for diagnostic or therapeutic purposes in patients with bile duct obstruction. Recently, cholangiography has been substituted by less invasive and accurate techniques, such as MRCP and MDCT scanning [52]. However, it is still useful for histological confirmation of biliary tract diseases or preoperative drainage. Many physicians and surgeons still rely on cholangiography rather than MRCP to determine the extent and location of biliary obstruction. Extrahepatic CCA shows longitudinal spread along the bile duct, often resulting in a residual tumor at the surgical margin. Preoperative assessment of the longitudinal spread of bile duct cancer has been conducted by mapping biopsy using percutaneous transhepatic cholangioscopy [59,60]. With the combination of percutaneous transhepatic cholangioscopy and cholangiography, its accuracy improved to 80–92% [60,61].

Whether to utilize PTC or ERCP is determined by the physician according to the location of the lesion or the underlying disease. Patients with PSC who are difficult to access percutaneously due to significant biliary stricture are optimal candidates for ERCP. Mapping biopsies up to the hilar bile duct and targeted biopsies of the biliary stricture can be taken using a novel device delivery system (EndoSheather; Piolax, Kanagawa, Japan) [62,63]. This device serves as a conduit for the biopsy forceps, avoiding repeated and direct contact with the duodenal papilla and the malignant biliary stricture, post-ERCP pancreatitis, and contamination with cancer cells is prevented.

Brush cytology or biopsy of the suspicious lesion is conducted through ERCP or PTC. In a meta-analysis of 1,123 patients with CCA, the diagnostic sensitivity was 56% with brushing, 67% with biopsy, and 70.7% with brushing and biopsy combined [64]. The combination of an elevated CA 19-9 and a positive brush cytology showed a sensitivity and specificity of 88% and 97%, respectively [28]. A case of extrahepatic CCA diagnosed by a biopsy during an ERCP is presented (Figure 1).

### 5.2. Endoscopic Ultrasound

EUS can assess the regional lymph node status and the local extent of extrahepatic CCA. Although cholangiography with or without brushings and biopsies is the primary choice for the diagnosis of CCA, EUS with fine needle aspiration (FNA) plays an important role in clinical practice. EUS-guided FNA can be performed to acquire tissue samples from tumors and enlarged lymph nodes [65]. EUS-guided FNA is more sensitive than ERCP with brushings for diagnosing extrahepatic CCA [66]. FNA minimizes the contamination of the biliary tree, which might occur during ERCP, allowing for the adequate acquisition of tissues. It has been shown that the sensitivity of EUS-FNA is influenced by the location of the tumor [66]. Several studies have reported successful uses of EUS-FNA for the diagnosis of extrahepatic CCA; the sensitivity for CCA diagnosis in these studies ranged from 43% to 89%, with the majority of studies having a sensitivity of over 70% [67,68,69,70,71]. However, EUS has a lower sensitivity for staging intrahepatic CCAs [71,72]. Figure 2 is a case of EUS-guided FNA and biopsy in patients with a suspected intrahepatic CCA.

### 5.3. Intraductal Ultrasound

Intraductal ultrasound (IDUS) is performed by inserting a high-frequency (12–30 MHz) ultrasound miniprobe into the biliary tract [73]. IDUS distinguishes malignant from benign strictures by recognizing specific sonographic imaging characteristics. The following ultrasound characteristics are suggestive of malignancy: eccentric wall thickening with an uneven surface, a hypoechoic mass, disruption of the normal three-layer, heterogeneity of the internal echo pattern, and a papillary surface [74]. Additionally, IDUS can improve the accuracy of the local staging of CCA. IDUS can be applied in the early detection of tumors, evaluation of longitudinal tumor extent, and detection of tumor invasion to surrounding organs or blood vessels [75]. IDUS is more effective than EUS for examining the proximal biliary system and surrounding structures, such as the portal vein and right hepatic artery. However, it is difficult to assess distant tissues or lymph nodes with IDUS because of its limited depth of penetration. Furthermore, IDUS cannot be used for FNA, unlike EUS.

### 5.4. Cholangioscopy

Cholangioscopy is a device that visualizes the biliary tract by inserting a thin-diameter cholangioscope directly into the bile duct [76]. Cholangioscopy can be performed via peroral or percutaneous transhepatic routes. The oral route is generally preferred because the percutaneous transhepatic technique carries a risk of metastases along the sinus tract or the peritoneum, as well as prolonged hospitalization and problems including hemorrhage and bile leak [59]. When compared to ERCP alone, cholangioscopy with and without biopsy is associated with an improved diagnostic yield. It can be used to analyze ambiguous fluoroscopy findings during ERCP, to evaluate the extent of CCA before surgery, and to find stones not visible in traditional cholangiography. Additionally, the cholangioscopy allows for direct visualization of the biliary epithelium, which can be utilized for targeted biopsies of bile duct lesions and lithotripsy [77,78]. When compared to that of cytology and ERCP-guided biopsy (34% and 54%, respectively), single-operator cholangioscopy with biopsy has demonstrated a potential, incremental accuracy as high as 85% for diagnosing indeterminate strictures [77,79,80]. Additionally, it allows passing endobiliary instruments to achieve endomicroscopy, laser/electrohydraulic lithotripsy, and basket retrieval. Figure 3 is a cholangioscopic finding of hilar CCA.

Table 1 shows the advantages and limitations of endoscopic modalities (ERCP, EUS, and cholangioscopy) used in the diagnosis of CCA.

## 6. Pathologic Findings

Histologically, intrahepatic CCAs are mostly adenocarcinomas with variable microscopic patterns. Small-duct intrahepatic CCAs often show a growth pattern resembling small biliary ductules. Large-duct intrahepatic and extrahepatic CCAs are similar, often characterized by a tubular pattern, desmoplastic stroma, and frequent perineural and lymphovascular invasion [6]. Immunohistochemistry can be helpful for the differential diagnosis of primary hepatic tumors [81]. Almost all CCAs show strong positivity for cytokeratin (CK)7 and CK19. Among common adenocarcinomas, CK7 positivity is consistent with biliary tract origin. However, metastatic cancers of the lung and breast are also CK7 positive, and the diagnosis of a CCA may be a diagnosis of exclusion. Positivity for CK20 can be seen in up to 20% of cases of intrahepatic CCA. An antibody panel consisting of hepatocyte paraffin 1, arginase-1, monoclonal carcinoembryonic antigen, CK7, CK20, TTF-1 (positive cytoplasmic staining in HCC, positive nuclear staining in lung adenocarcinoma), and CDX2 (positive nuclear staining in intestinal adenocarcinoma) could be used to optimize the differential diagnosis of HCC, metastatic adenocarcinoma, and CCA [82]. It is clinically important to rule out HCC, which is variably positive for hepatocyte paraffin 1, glypican-3, and arginase-1.

## 7. Genomic Heterogeneity

Molecular alterations affecting the tumorigenesis of CCA have been defined. The most prevalent genetic alterations identified in CCA affect key networks such as DNA repair (TP53) [83,84], the WNT–CTNNB1 pathway [85], tyrosine kinase signaling (KRAS, BRAF, SMAD4, and FGFR2) [86,87,88], protein tyrosine phosphatase (PTPN3) [89], epigenetic (IDH1 and IDH2) [83,84,90,91] and chromatin-remodeling factors (histone-lysine N-methyltransferase 2C, also known as MLL3) [86], including the SWI/SNF complex (ARID1A, PBRM1, and BAP1) [83,84,90,91] and deregulated Notch signaling, which is a key component in cholangiocyte differentiation and biliary duct development.

## 8. Summary

CCA refers to malignancies of the perihilar, extrahepatic, or intrahepatic bile duct. Most patients with extrahepatic or perihilar CCA present asymptomatic jaundice, weight loss, and right upper quadrant discomfort. However, patients with intrahepatic CCA seldom have jaundice.

Cross-sectional scans, such as MDCT or MRI with MRCP, are frequently used for initial examination. Extrahepatic CCA shows abrupt changes in ductal diameter with intrahepatic and extrahepatic ductal dilation. In perihilar CCA, intrahepatic ducts are dilated, but extrahepatic ducts remain normal in size. Intrahepatic CCA is usually detected as a mass lesion within a non-cirrhotic liver.

All patients with suspected CCA should be checked for tumor biomarkers (CA 19-9, CEA), and patients with intrahepatic tumors should additionally undergo an AFP check-up. Tumor biomarkers can be used not only for the diagnosis of CCA but also for treatment monitoring and determining recurrence.

For extrahepatic CCA, EUS or ERCP is recommended as the next step in the evaluation process. Tissue can be obtained through EUS-FNA or ERCP, and therapeutic intervention, such as stent insertion, can be conducted via ERCP. In the case of perihilar lesions, MDCT or MRCP is performed first, followed by ERCP for histological confirmation. In patients with intrahepatic lesions, CCA and HCC should be differentiated through cross-sectional imaging and measuring tumor markers. For surgical candidates without extra-regional lymph node involvement or distant metastases, a PET scan is recommended.

## Figures and Tables

**Figure 1 diagnostics-13-00233-f001:**
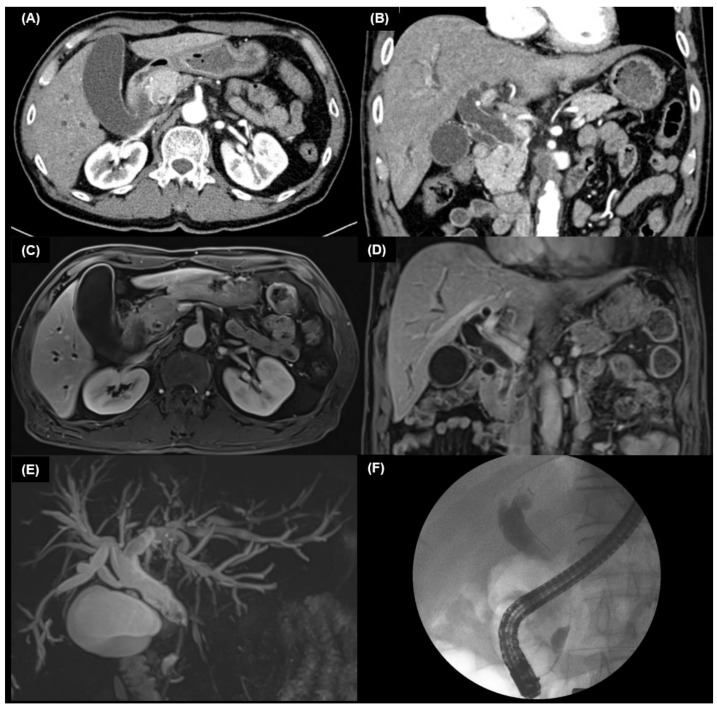
Images of a 64-year-old man with extrahepatic cholangiocarcinoma. Transverse (**A**) and coronal (**B**) image of arterial phase contrast-enhanced computed tomography shows approximately 2 cm length segmental wall thickening and enhancement of intrapancreatic common bile duct causing dilatation of upstream bile ducts. (**C**,**D**) Axial T1-weighted image: a solid mass with minimal contrast enhancement in the distal common bile duct. (**E**) Three-dimensional magnetic resonance cholangiopancreatography (MRCP) image demonstrates the segmental obstruction of the distal common bile duct. (**F**) endoscopic retrograde cholangiopancreatography (ERCP) shows distal common bile duct stricture with proximal duct dilatation.

**Figure 2 diagnostics-13-00233-f002:**
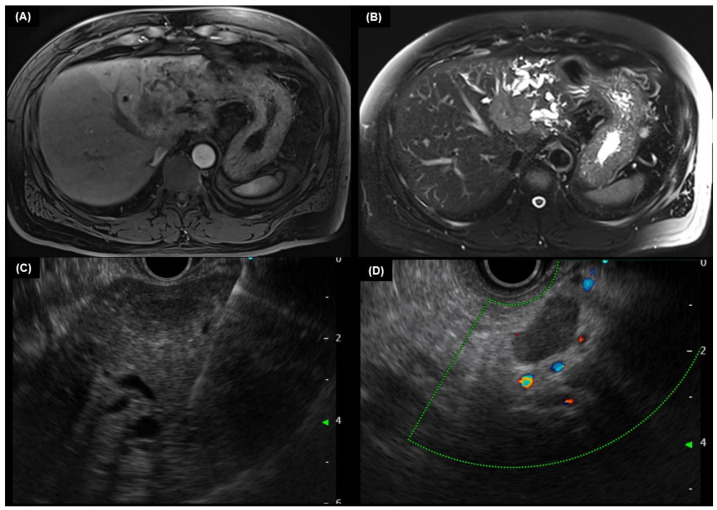
A 45-year-old male patient diagnosed with mass-forming intrahepatic cholangiocarcinoma. T1-weighted (**A**) and T2-weighted image (**B**) of magnetic resonance imaging showed a 4.7 cm targetoid rim arterial enhancing mass with intrahepatic bile duct dilatation and separation. An endoscopic ultrasound-guided fine needle aspiration and biopsy was performed on the hepatic mass (**C**) and the adjacent enlarged lymph node (**D**). The pathologic result was adenocarcinoma.

**Figure 3 diagnostics-13-00233-f003:**
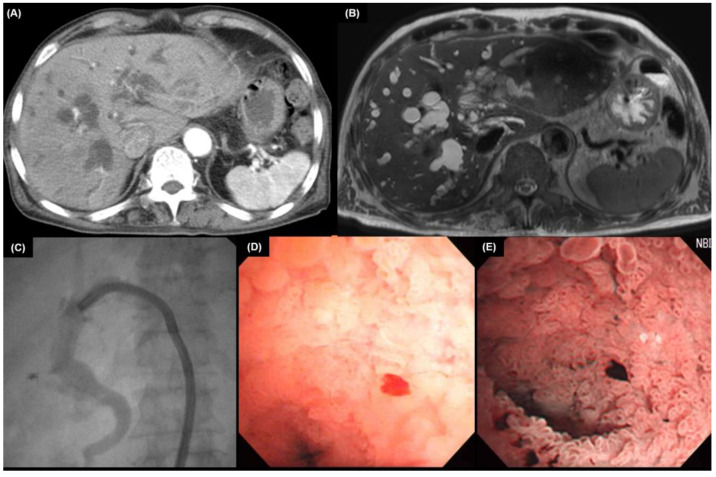
Computed tomography (**A**) and magnetic resonance imaging (**B**) of an 85-year-old male patient showed a hilar bile duct obstruction and dilated intrahepatic bile ducts. Choledochoscope was inserted, (**C**) and infiltrating type hilar cholangiocarcinoma (**D**) was observed. (**E**) Granular mucosa in narrow band imaging is observed.

**Table 1 diagnostics-13-00233-t001:** Advantages and limitations of endoscopic modalities for diagnosing cholangiocarcinoma.

	Advantages	Limitations
ERCP	Anatomic delineation for characterizing biliary stricture Real-time visualization facilitating biopsy or stenting	Invasive Incomplete evaluation for proximal ducts in high-grade biliary obstruction
EUS	Allows detailed examination of the extrahepatic bile duct and surrounding structures (lymph nodes, vessels) EUS-FNA/B enables cytologic/pathologic examination	Rarely, the risk of tumor seeding along the needle tract during EUS-FNA/B
SOC	Direct visualization of the biliary epithelium Allows targeted biopsy	Technically challenging (widely opened AoV orifice with a major sphincterotomy is needed)

Abbreviations: ERCP, Endoscopic retrograde cholangiopancreatography; EUS, Endoscopic ultrasound; FNA/B, Fine needle aspiration/biopsy; SOC, Single-operator cholangioscopy; AoV, ampulla of Vater.

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
