# Peer review of "Diagnosis of Cholangiocarcinoma"

_diagnostics, 2023, doi:10.3390/diagnostics13020233_

Round 1

Reviewer 1 Report

The paper is well written. If I give some comments, the role of step biopsy of the bile duct via endoscope for evaluating the extent of cancer should be described.

Author Response

December 31st, 2022

RE: diagnostics-2102976, entitled " Diagnosis of Cholangiocarcinoma" coauthored by Dong Woo Shin, Sung-Hoon Moon.

Thank you very much for giving us an opportunity for revision.
Accurate and kind comments by the reviewer have been addressed in the discussion. We also believe that these comments improved our manuscript. Changes have been made by changing the color to Red in the revised manuscript to avoid any confusion. 

I anticipate good response.
Thank you!

Sincerely,
Jong Hyeok Kim, M.D., Ph.D.
Dong Woo Shin, M.D.

Reply to Reviewer’s comments
Question #1. The paper is well written. If I give some comments, the role of step biopsy of the bile duct via endoscope for evaluating the extent of cancer should be described.
Author’s comments: Thank you for your kind comment. Accurate diagnosis of the extent of the cholangiocarcinoma is essential to enable selection of the appropriate medical and surgical therapy. Preoperative endoscopic assessment of longitudinal spread of bile duct cancer was described as follows: (Page 6 of 17, line 249-253) ‘Extrahepatic CCA shows longitudinal spread along the bile duct, often resulting in residual tumor at the surgical margin. Preoperative assessment of longitudinal spread of bile duct cancer has been conducted by mapping biopsy using percutaneous transhepatic cholangioscopy [59, 60]. With the combination of percutaneous transhepatic cholangioscopy and cholangiography, its accuracy improved to 80-92% [60, 61].’

The authors really appreciated the reviewer’s kind and accurate comments. The revision based on these comments made this manuscript more accurate and the quality improved. Thank you again. 

Jong Hyeok Kim, M.D., Ph.D.
Dong Woo Shin, M.D.

Reviewer 2 Report

Dr. Shin et al provide a review of how to diagnose cholangiocarcinoma. It is thorough and can be helpful for non-experts. My comments are as follows.

1.       Risk factors can be a useful addition to this manuscript. In addition to chronic liver disease, congenital anomalies, liver flukes and carcinogens may be mentioned.

2.       A review of diagnosis of cholangiocarcinoma is not complete without a discussion of pathology. Immunohistochemistry becomes important when conducting liver biopsies for suspected intrahepatic cholangiocarcinoma, as the lesion must be distinguished from HCC and metastases.

3.       Discussion of endoscopic methods such as biopsy using endosheaths may be helpful, but is not essential.

4.       Recent advances such as genomic profiling (liquid biopsy, determining subtypes for treatment such as FGFR2 fusion, HER2, etc.) may also be good additions.

Author Response

December 31st, 2022

RE: diagnostics-2102976, entitled " Diagnosis of Cholangiocarcinoma" coauthored by Dong Woo Shin, Sung-Hoon Moon.

Thank you very much for giving us an opportunity for revision.
Accurate and kind comments by the reviewer have been addressed in the discussion. We also believe that these comments improved our manuscript. Changes have been made by changing the color to Red in the revised manuscript to avoid any confusion. 

I anticipate good response.
Thank you!

Sincerely,
Jong Hyeok Kim, M.D., Ph.D.
Dong Woo Shin, M.D.

Reply to Reviewer’s comments
Dr. Shin et al provide a review of how to diagnose cholangiocarcinoma. It is thorough and can be helpful for non-experts. My comments are as follows.

Question #1. Risk factors can be a useful addition to this manuscript. In addition to chronic liver disease, congenital anomalies, liver flukes and carcinogens may be mentioned.
Author’s comments: Thank you for your kind comment.  We described in the second paragraph of the introduction as follows: (Page 1-2 of 17, line 42-46) ‘The established risk factors for CCA include primary sclerosing cholangitis, parasite infection, biliary-duct cysts, and hepatolithiasis. Less-established risk factors include inflammatory bowel disease, chronic liver disease (cirrhosis, chronic hepatitis B and C), obesity, diabetes, alcohol, smoking, and genetic polymorphisms [6, 9].’

Question #2. A review of diagnosis of cholangiocarcinoma is not complete without a discussion of pathology. Immunohistochemistry becomes important when conducting liver biopsies for suspected intrahepatic cholangiocarcinoma, as the lesion must be distinguished from HCC and metastases.
Author’s comments: Thank you for your important comment. We described the pathology and immunohistochemistry of cholangiocarcinoma as follows: (Page 10 of 17, line 346-362) ‘6. PATHOLOGIC FINDINGS Histologically, intrahepatic CCAs are mostly adenocarcinomas with variable micro-scopic patterns. Small-duct intrahepatic CCAs often show a growth pattern resembling small biliary ductules. Large-duct intrahepatic and sxtrahepatic CCAs are similar, often characterized by a tubular pattern, desmoplastic stroma, and frequent perineural and lymphovascular invasion [6]. Immunohistochemistry can be very helpful for the dif-ferential diagnosis of primary hepatic tumors [81]. Almost all CCAs show strong posi-tivity for cytokeratin (CK)7 and CK19. Among common adenocarcinomas, CK7 posi-tivity is consistent with biliary tract origin. However, metastatic cancers of the lung and breast are also CK7 positive, and the diagnosis of a CCA may be a diagnosis of exclusion. Positivity for CK20 can be seen in up to 20% of cases of intrahepatic CCA. An antibody panel consisting of hepatocyte paraffin 1, arginase-1, monoclonal carcinoembryonic antigen, CK7, CK20, TTF-1 (positive cytoplasmic staining in HCC, positive nuclear staining in lung adenocarcinoma), and CDX2 (positive nuclear staining in intestinal adenocarcinoma) could be used to optimize the differential diagnosis of HCC, meta-static adenocarcinoma, and CCA [82]. It is clinically important to rule out HCC, which is variably positive for hepatocyte paraffin 1, glypican-3, and arginase-1.’

Question #3. Discussion of endoscopic methods such as biopsy using endosheaths may be helpful, but is not essential.
Author’s comments: Thank you for your comment. The mapping biopsy using EndoSheather was described as follows: (Page 6 of 17m line 257-261) ‘Mapping biopsies up to the hilar bile duct and targeted biopsies of the biliary stricture can be taken using a novel device delivery system (EndoSheather; Piolax, Japan) [62, 63]. This device serves as a conduit for the biopsy forceps, avoiding repeated and direct contact with the duodenal papilla and the malignant biliary stricture, post-ERCP pan-creatitis and contamination with cancer cells is prevented.’

Question #4. Recent advances such as genomic profiling (liquid biopsy, determining subtypes for treatment such as FGFR2 fusion, HER2, etc.) may also be good additions.
Author’s comments: Thank you for your comment. We describe the genomic profiling of CCA as follows: (Page 10 of 17, line 364-372) ‘Molecular alterations affecting the tumorigenesis of CCA have been defined. The most prevalent genetic alterations identified in CCA affect key networks such as DNA repair (TP53) [83, 84], the WNT–CTNNB1 pathway [85], tyrosine kinase signalling (KRAS, BRAF, SMAD4 and FGFR2) [86-88], protein tyrosine phosphatase (PTPN3) [89], epigenetic (IDH1 and IDH2) [83, 84, 90, 91] and chromatin-remodelling factors (his-tone-lysine N-methyltransferase 2C, also known as MLL3) [86], including the SWI/SNF complex (ARID1A, PBRM1 and BAP1) [83, 84, 90, 91] and deregulated Notch signalling, which is a key component in cholangiocyte differentiation and biliary duct development.’

The authors really appreciated the reviewer’s kind and accurate comments. The revision based on these comments made this manuscript more accurate and the quality improved. Thank you again. 

Jong Hyeok Kim, M.D., Ph.D.
Dong Woo Shin, M.D.

Round 2

Reviewer 2 Report

The authors have adequately revised their manuscript. I would like to congratulate the authors for their fine work. I have no further comments.